# Genomic Characterisation of an Isolate of *Brassica Yellows Virus* Associated with Brassica Weed in Tasmania

**DOI:** 10.3390/plants11070884

**Published:** 2022-03-25

**Authors:** Muhammad Umar, Tahir Farooq, Robert S. Tegg, Tamilarasan Thangavel, Calum R. Wilson

**Affiliations:** 1New Town Research Laboratories, Tasmanian Institute of Agriculture, University of Tasmania, 13 St. Johns Avenue, New Town, TAS 7008, Australia; m.umar@utas.edu.au (M.U.); robert.tegg@utas.edu.au (R.S.T.); tamil.thangavel@daf.qld.gov.au (T.T.); 2Guangdong Provincial Key Laboratory of High Technology for Plant Protection, Plant Protection Research Institute, Guangdong Academy of Agricultural Sciences, Guangzhou 510640, China; tfarooq@gdppri.com; 3Department of Agriculture and Fisheries (Queensland), Bundaberg Research Facility, 49 Ashfield Road, Bundaberg, QLD 4670, Australia

**Keywords:** *Brassica yellows virus*, genome characterisation, recombination, *Polerovirus*, phylogenetic analysis

## Abstract

*Brassica yellows virus* (BrYV), a tentative species in the genus *Polerovirus*, of the *Solemoviridae* family, is a phloem-restricted and aphid-transmitted virus with at least three genotypes (A, B, and C). It has been found across mainland China, South Korea, and Japan. BrYV was previously undescribed in Tasmania, and its genetic variability in the state remains unknown. Here, we describe a near-complete genome sequence of BrYV (genotype A) isolated from *Raphanus raphanistrum* in Tasmania using next-generation sequencing and sanger sequencing of RT-PCR products. BrYV-Tas (GenBank Accession no. OM469309) possesses a genome of 5516 nucleotides (nt) and shares higher sequence identity (about 90%) with other BrYV isolates. Phylogenetic analyses showed variability in the clustering patterns of the individual genes of BrYV-Tas. Recombination analysis revealed beginning and ending breakpoints at nucleotide positions 1922 to 5234 nt, with the BrYV isolate LC428359 and BrYV isolate KY310572 identified as major and minor parents, respectively. Results of the evolutionary analysis showed that the majority of the codons for each gene are evolving under purifying selection, though a few codons were also detected to have positive selection pressure. Taken together, our findings will facilitate an understanding of the evolutionary dynamics and genetic diversity of BrYV.

## 1. Introduction

The genus *Polerovirus* (family *Solemoviridae*) encompasses ~32 species (including both tentative and approved) of genetically diverse, and economically important plant viruses [1]. *Brassica yellows virus* (BrYV) is a tentative polerovirus that was first reported infecting brassica spp. in China in 2011 [2]. Since then, it has been identified in other countries, including South Korea in 2015 [3], Japan in 2019 [4], and Australia in 2021 [5]. Similar to other poleroviruses, BrYV is a phloem-limited virus that is transmitted by specific aphid species in a persistent circulative manner. Several cruciferous plant species may be naturally infected by BrYV, including oilseed rape (*Brassica napus*), Chinese cabbage (*B*. *chinensis*), cabbage (*B*. *oleracea*), cauliflower (*B*. *oleracea var*. *botrytis*), rutabaga (*B*. *napobrassica*), leaf mustard (*B*. *juncea*), radish (*Raphanus sativus var*. *oleifera*), and white glabrous mustard (*B*. *alboglabara*). BrYV infection in crucifer plants induces mottling, yellowing, and leaf malformation symptoms [2,3].

The full-length genome of BrYV consists of a 5.6–5.7 kb long positive-sense single-stranded RNA (+ssRNA) which contains six open reading frames (ORFs), a short untranslated region (UTR) at 5’, an intergenic non-coding region (NCR) spanning over ORF2 and ORF3, and a UTR at the 3’ end that lacks a poly (A) tail or tRNA-like structure. The genomic RNA is packed into a spherical-shaped virion of 25–30 nm diameter. Three unique genotypes, BrYV-A, BrYV-B, and BrYV-C, with genome sizes of 5666, 5666, and 5678 nt, respectively, have been described [2,6]. Additionally, a recombinant strain (BrYV-CS) has been reported in Chinese cabbage in South Korea [3]. Like typical poleroviruses, ORFs 0–2 are translated from genomic RNA, while ORFs 3–5 are translated from sub-genomic RNA (Figure 1). ORF0 expresses the P0 protein that functions as a determinant of host range and symptoms and acts as a silencing suppressor. The fusion of ORF1 and ORF2 encodes the P1–2 fusion protein or replication-associated protein (RepA). ORF3 encodes the major P3 or coat protein (CP) which sits in-frame with ORF5. The P3–5 overlapped translation results in the expression of a read-through protein (P5) which resides on the external surface of the mature virion and is associated with the long-distance movement and aphid transmission of the virus [7]. The ORF4 encodes the P4 protein, which plays an important role in virus systemic movement within infected plants (Figure 1).

A recent study in Australia demonstrated that the highest genomic variability in BrYV, and its closely relative *Turnip yellows virus* (TuYV), occured in ORF5 [5]. Based on phylogenetic and recombination analyses, this study proposed that BrYV and TuYV could be regarded as variants of the same species. However, a number of other studies have examined the genetic variation across other ORFs which differentiate BrYV from TuYV [2,3,6].

RNA recombination predominantly drives genetic diversity and plays a crucial role in the emergence, evolution, and epidemiology of the plant-infecting RNA viruses. It appears to be the main mechanism behind the formation of new and more pathogenic viral strains/species [8,9]. The genera *Polerovirus* and *Enamovirus* of the *Solemoviridae* family (previously in the family *Luteoviridae*) [10] are known to be commonly associated with recombination events [9]. Based on the phylogenetic analysis, several studies have demonstrated that self-recombination among poleroviruses or recombination with other luteoviruses has led to the observed evolvution of poleroviruses [11,12,13].

Recently, complete genome sequencing of different plant viruses has expanded our understanding of their molecular characterisation, genetic diversity, and evolutionary relationships [14,15,16]. Here, we took advantage of next-generation sequencing (NGS) to obtain a near-complete genome sequence of a BrYV isolate infecting *R. raphanistrum* in Tasmania. We employed RT-PCR and multiplex RT-PCR to fill sequence gaps and confirm the BrYV genotype, respectively. Furthermore, we performed phylogenetic and recombination detection analyses to improve our understanding of the existing genetic variability and evolutionary relatedness between the newly described isolate and other globally reported BrYV isolates. The findings of our study will provide valuable information regarding the genomic diversity and recombinational dynamics of BrYV to better understand and manage plant virus diseases.

## 2. Results

### 2.1. NGS Analysis

In total, 22,351,000 raw reads were obtained with Illumina sequencing for the tested sample. After quality trimming using BBDuK, these numbers were reduced to 22,228,228. De novo assembly generated sixteen contigs (ranged 3995–121 bp with N50 = 511 bp), produced by 670 viral reads. A BLASTn/BLASTx search of the GenBank database found a viral contig of ~3995 bp in sizes corresponding to the BrYV isolate “TO3” (Accession no. LC428360), sharing 96.68% nucleotide similarity.

### 2.2. Validation of NGS Results and Amplification of Missing Genome Sequence

Since the tested sample was a composite from five different plants, this necessitated further testing of individual leaves to determine the identification of the virus source. BrYV infection was confirmed in one *R. raphanistrum* sample by multiplex RT-PCR. Multiplex RT-PCR gave a amplicon of 277 bp which also revealed that the BrYV-Tas isolate belongs to genotype-A. To amplify the missing parts of the BrYV-Tas genome (i.e., 5′UTR, P0, and part of P1-P2 towards 5′ end), RT-PCR assays were conducted with primers BrY-Ntab001F (+)/TuYVOrf0R and primer BrY-Ntab724F(+)/BrY-Ntab2064R(-) (Table 1), using cycling conditions, as previously described [17] with products sequenced. Consequently, the entire genome sequence of the BrYV Tasmanian isolate was determined, except for 138 bp at the termini of the 3′UTR (GenBank Accession no. OM469309).

### 2.3. Multiple Sequence Alignment and Phylogenetic Analysis

The genomic sequence obtained for BrYV-Tas was 5516 nt in length and had a typical BrYV genomic organisation, with six ORFs (ORF0−ORF5), a 31 nt 5′-UTR, and an incomplete 47 nt at 3′-UTR. ORF0 spans nt 32 to 781 and is predicted to encode the P0 suppressor. ORF1 stretches between nt 174 to nt 1967 and encodes the P1 protein. ORF2 (nt 1853–3268) is assumed to be translated into a P1-P2 fusion protein (RdRp) via a −1 ribosomal frameshift. GGGAAAG at nt 1617 to 1623 is the shifty sequence. ORF3 starts at nt 3460 to 4068, encoding a putative coat protein (CP). ORF4 starts at nt 3491 to 4018 and codes for a putative P4 protein by a leaky scanning. ORF5 (from nt 4255 to 5469) is translated into the P3-P5 fusion protein via a read-through strategy, which is thought to play a key role in virus transmission (Figure 1).

Sequence homology and phylogenetic analysis based on the full genome indicated that the BrYV-Tas shares higher sequence identities (94.4%) with two Japanese BrYV isolates LC428360 and LC428361 and they cluster together in the same clade (II) (Figure 2A) (Appendix A). These Japanese isolates were found from *B. rapa* and *B. napus* hosts, respectively. The lowest homology percentage (81.3%) was observed for an isolate from China (MK057527) infecting tobacco. BrYV-Tas shared only 78.2% of nucleotide similarity with the closest TuYV isolate genomic sequence available on GenBank (NC_003743) (Appendix A).

Subsequent alignments indicated that the P0, P1, P1-P2, P3, P3-P5, P4, and P5 genes of BrYV-Tas shared sequence identities of 90.9–98.1% for P0 (Appendix A), 87.1–96.6% for P1 (Appendix A), 89.3–92.1% for P1-P2 (Appendix A), 92.6–98.9% for P3 (Appendix A), 75.3–98.5% for P3-P5 (Appendix A), 92.0–99.1% for P4 (Appendix A), 67.1–98.8% for P5 (Appendix A) with the corresponding genes of other BrYV isolates examined. Notably, phylogenetic analysis revealed that the clustering pattern of the individual BrYV-Tas genes, mentioned above, was variable (Figure 2). The percentage of nucleotide sequence identities between BrYV-Tas and TuYV (NC_003743) was 88.1% for P0 (Appendix A), 86.6% for P1 (Appendix A), 88.9% for P1-P2 (Appendix A), 95.2% for P3 (Appendix A), 67.0% for P3-P5 (Appendix A), 95.5% for P4 (Appendix A), and 55.2% for P5 (Appendix A). Meanwhile, the amino acid percentage identities between BrYV-Tas and TuYV (NC_003743) was 82.7% for P0 (Appendix A), 85.0% for P1 (Appendix A), 88.9% for P1-P2 (Appendix A), 95.0% for P3 (Appendix A), 60.9% for P3-P5 (Appendix A), 90.9% for P4 (Appendix A), and 43.2% for P5 (Appendix A).

**Figure 2 plants-11-00884-f002:**
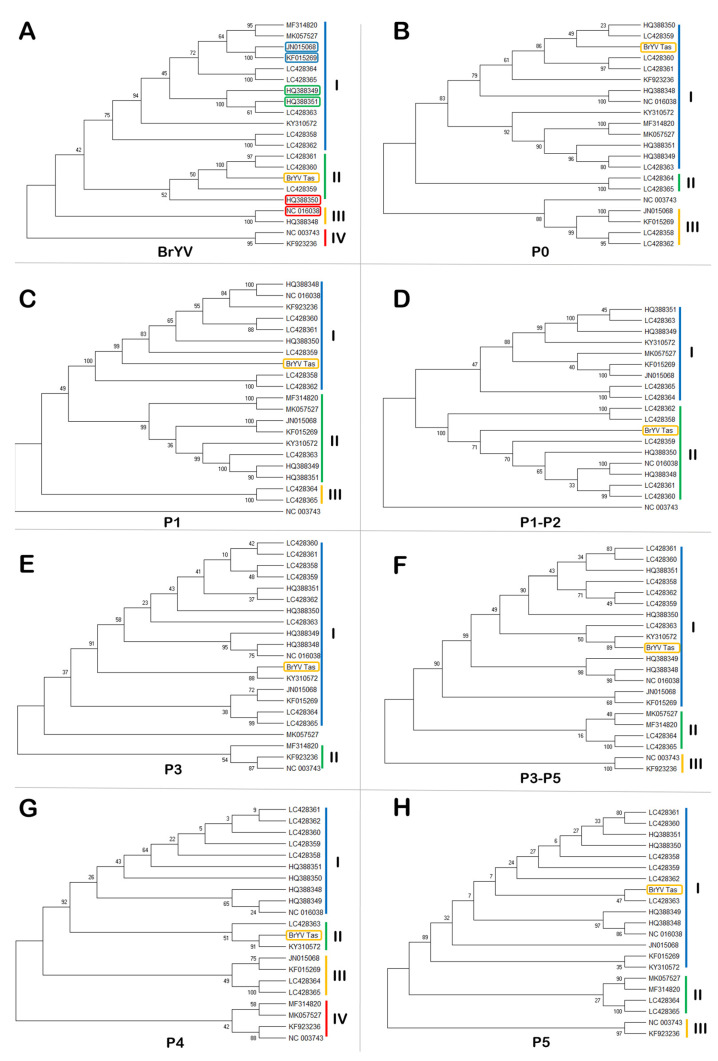
Phylogenetic trees (maximum likelihood algorithm) based on nucleotide sequence alignments of *Brassica Yellows Virus* (BrYV) constructed in MEGA-X. Bootstrap values shown at the nodes indicate the percentage of 1000 replications supporting the branching patterns shown. Phylogenetic analysis includes full/near-full genomes (**A**) and each gene responsible to encode their respective proteins (**B**) P0, (**C**) P1, (**D**) P1-P2, (**E**) P3, (**F**) P3-P5, (**G**) P4, and (**H**) P5. The accessions with red, green, and blue rectangles represent BrYV genotypes A, B, and C, respectively. Yellow rectangles indicate the BrYV-Tas isolate from the present study. Additional information regarding all these isolates is given in Table 2.

**Table 2 plants-11-00884-t002:** BrYV and TuYV reference isolates retrieved from GenBank.

Isolate	Accession No.	Host	Country	References
BrYV-Tas	OM469309	*Raphanus raphanistrum*	Australia, Hobart	This study
BrYV-ABJ	HQ388348	*Brassica napus* *var.* *napobrassica*	China: Beijing	[2]
BrYV-BBJ	HQ388349	*B. napus* *var.* *napobrassica*	China: Beijing	[2]
BrYV-AJS	HQ388350	*B. campestris* L.	China: Jiangsu	[2]
BrYV-BJS	HQ388351	*B. campestris* L.	China: Jiangsu	[2]
BrYV-CR	JN015068	*R. raphanistrum*	China: Haidian	[6]
BrYV-CC	KF015269	*B. rapa pekinensis*	China: Beijing	[6]
BrYV-CS	KF923236	*B.rapa*	South Korea: Cheongsong	[3]
BrYV-China	KY310572	*B. napus*	China	Direct submission
BrYV-CC1	LC428358	*B. rapa* subsp. *pekinensis*	Japan: Hokkaido	[4]
BrYV-WN1	LC428359	*Sinapis alba*	Japan: Hokkaido	[4]
BrYV-TO3	LC428360	*B. rapa* subsp. *Rapa*	Japan: Okayama	[4]
BrYV-NAP	LC428361	*B. napus*	Japan: Okayama	[4]
BrYV-CD9	LC428362	*B. oleracea* *var.* *capitata*	Japan: Hokkaido	[4]
BrYV-R3b	LC428363	*R. sativus*	Japan: Okayama	[4]
BrYV-RT8	LC428364	*R. sativus*	Japan: Hokkaido	[4]
BrYV-R40	LC428365	*R. sativus*	Japan: Okayama	[4]
BrYV-Anhui	MF314820	*Nicotiana tabacum*	China	Direct submission
BrYV-NtabQJ	MK057527	*N. tabacum*	China: Yunnan	[17]
BrYV-ABJ	NC_016038	*B. napus* *var.* *napobrassica*	China: Beijing	[2]
TuYV-FL1	NC_003743	*Lactuca sativa*	France: Avignon	[20]

### 2.4. Recombination Analysis

When the full genomic sequences of different BrYVs were aligned, a significant recombination event for BrYV-Tas (recombinant), with LC428359 (major parent) and KY310572 (minor parent), was strongly supported using all seven methods in the RDP4 package, including RDP, GENECONV, BootScan, MaxChi, Chimaera, SiScan, and 3Seq (Figure 3A). The *p*-values for these seven methods were as follows: RDP (3.103 × 10^−11^), GENECONV (2.109 × 10^−12^), BootScan (5.923 × 10^−12^), MaxChi (1.002 × 10^−15^), Chimaera (3.518 × 10^−16^), SiScan (2.511 × 10^−27^), and 3Seq (3.068 × 10^−19^). We observed that the recombinant region covered almost 41% of the BrYV genome, and the predicted beginning and ending breakpoints are located at nt 1922 in the region of P1-P2 and nt 5234 in the P5 region, respectively (Figure 3B).

### 2.5. Nucleotide Diversity and Haplotype Variability Indices

We analysed all datasets comprising BrYV full genomes and individual genes to compare the standing molecular diversity among viral populations arising from different geographical locations. The diversity determined for the whole genome analysis of BrYV was S = 1477, Eta = 1684, π = 0.06883, H_d_ = 0.9950, *θw* = 0.07683, Tajima’s *D* = −0.89807 (Figure 4A and Table 3).

Genetic diversity analysis for BrYV individual genes also revealed that all genes were variable with high numbers of polymorphisms and polymorphic sites, very high haplotype diversity, but low nucleotide diversity (Figure 4A and Table 3). Aside from P3 (S = 87, Eta = 94, π = 0.03297, h = 0.989) and P4 (S = 72, Eta = 78, π = 0.03147, H_d_ = 0.932), each gene indicated high diversity and had a high proportion of haplotypes. P5 with Eta = 578, π = 0.07522, and H_d_ = 0.995 appeared to be the most genetically diverse gene that is three times more diverse than that of the conserved P3/P4 region of the BrYV genome (Table 3). Following P5, P1 (S = 451, Eta = 528, π = 0.08736, H_d_ = 0.995) appeared to be the gene with next greatest diversity (possibly linked to the genetically diverse region within P1) surpassing P0 (S = 159, Eta = 171, π = 0.06490, H_d_ = 0.995), previously regarded as the next most variable gene. Despite the fact that the polymorphic sites varied between 14 and 38% for each gene sequence, haplotype diversity was high, ranging from 93 to 100%, indicating that the number of unique sequences with different combinations of these polymorphic sites was high, resulting in greater diversity within the BrYV population (Table 3). Likewise, we also compared the genetic diversity between two genes encoding fusion proteins i.e., P1-P2 and P3-P5. Despite the existence of discrepancies between gene sizes, we were able to calculate average pairwise nucleotide diversities (π) for the aforementioned datasets (Figure 4A and Table 3).

The average number of segregating sites (*θw*) was greater for P5 (*θw* = 0.1307), followed by the P3-P5 (*θw* = 0.1006), the P1 (*θw* = 0.0704), the P1-P2 (*θw* = 0.0614), the P0 (*θw* = 0.0598), the P3 (*θw* = 0.0403), and the P4 (*θw* = 0.0384), respectively (Figure 4B and Table 3). The segregation rate exhibited by the overall BrYV population was *θw* = 0.0768. We also determined Tajima’s *D* values in the BrYV populations and individual genes which appeared to be negative, ranging from −2.075 to −0.251, indicating the presence of excessive polymorphic sites among these isolates (Figure 4C and Table 3). Although the P0 and P1 displayed slightly positive Tajima’s *D* values (0.0408 and 0.2512, respectively), this observation remained statistically non-significant.

### 2.6. Detection of Positive and Negative Selection Sites

We compared the non-synonymous to synonymous substitutions (d_N_/d_S_) for each gene to better understand the role of selection pressure on the genomic variation observed between the analysed datasets. DataMonkey-based analysis indicated that the genome of the BrYV population is being driven by purifying (negative) selection as the average dN/dS ratio of the BrYV genome remained <1 i.e., d_N_/d_S_ = 0.902 (Figure 5 and Table 4).

Overall, 1167/1848 sites were detected with negative selection, with values ranging between 0.051 and 0.949. Meanwhile, the proportion of positively selected sites remained lower (501/1848) than the negatively selected sites. Likewise, all genes displayed a variable frequency of negative selection with the lowest average d_N_/d_S_ value for P5 (0.371) and the highest average value for P1 (0.414) (Figure 5 and Table 4). A total of 353/425 sites were found under negative selection in the P5 gene sequence with values ranging from 0.050 to 0.949. Positive selection was detected at a few sites (25/425) with values ranging between 1.058 and 2.778; however, the proportion remained much lower than for negatively selected sites (Figure 6 and Table 4).

P1, with 282/607 negatively selected sites, displayed a similar pattern, having the average negative selection value of 0.319, with the lowest value of 0.050, and the highest of 0.942. Meanwhile, 22 sites were found to be positively selected in P1. With an average positive selection value of 1.548, the minimum value remained 1.063 and the maximum value was 4.401 (Table 4). P0 exhibited only nine positively selected sites with a value of 2.040, while this gene showed 201/249 negatively selected sites with 0.054 and 0.935 being the minimum and maximum values, respectively. P3 and P4 exhibited 11 positively selected sites, each with the ratio of the negatively selected sites being 122/202 and 127/176 for the P3 and P4 genes, respectively (Figure 6 and Table 4).

## 3. Discussion

The present study identified and genetically characterised an isolate of BrYV found infecting a *R. raphanistrum* host plant in Tasmania. Subsequent NGS data analysis revealed that this isolate of BrYV displayed a high percentage (96.68%) of nucleotide similarity with a BrYV isolate reported from Japan (LC428360). The NGS contig covered almost 70% of the genome of the BrYV reference sequence (NC_016038). A reconstructed genome, generated using NGS and RT-PCR, was 5516 bp long and had the typical organisation of BrYV. However, it was not possible to obtain the ~138 bp of the 3′ end of the genome, which includes the latter half of 3′ UTR. BrYV isolates from China, Korea, and Japan [2,3,6,17,21] have shown genetic variation in their ORFs and have been subdivided further into three genotypes (BrYV-A, B, and C) according to sequence comparisons and phylogenetic analysis [2,6]. This suggests that BrYV genomes, like those of other poleroviruses, are more diverse within different regions of the genome than previously thought, and that more diversity can be discovered after analysing whole-genome sequences. Our findings corroborate these previous studies [19], and multiplex RT-PCR analysis revealed that the BrYV-Tas isolate belongs to the BrYV genotype-A. Further analyses involving full genome and individual genes sequences of BrYV-Tas and other BrYV and TuYV isolates revealed that the presence of variation and clustering patterns were variable among different genes.

It has recently been proposed that TuYV and BrYV should be considered members of the same species if they share at least 83% nucleotide sequence identity [5]. The authors suggested a species demarcation criteria for new poleroviruses should include less than 83% nt sequence identity across all entire coding regions of the genome, as well as any significant differences in the host range, vector, and serology. The RdRp and CP regions were proposed as important for species demarcation with an 89% aa sequence identity in both regions for the members of the same polerovirus species [22], and CP is the most conserved region within poleroviruses [23,24]. However, currently, the working criterion for the demarcation of polerovirus species remains unchanged in the most recent report of ICTV on virus taxonomy [10], which is based on differences in host range, failure to cross-protect, differences in serological reactions, and differences in the amino acid sequence of any gene product greater than 10% [10,25]. Our findings are consistent with the recent species demarcation standards by ICTV [10] as well as with the previous studies related to characterisation of BrYV [2,3,6], as all genes of BrYV-Tas isolates shared relatively low (less than 90%) amino acid similarity with TuYV (NC_003743) with the exception of P3 and P4 genes which shared over 90% amino acid similarity with both BrYV and TuYV (Appendix A). Notably, for BrYV-Tas the cistrons at the 3′ end shared the least sequence identity (60.9% for P3-P5, and 43.2% for P5) with TuYV. Such difference in the P5 sequence between the BrYV and the TuYV isolates could be linked to variability in aphid transmission specificity or efficiency [26]. Based on the ICTV report, nucleotide sequence analysis, and phylogeny results our findings provide compelling evidence that BrYV-Tas is distinct from TuYV.

We have shown evidence of recombination in the genome of BrYV-Tas, with the analysis suggesting that the recombinant consisted of genomic parts from distinct BrYV isolates. Recombination events in RNA viruses have been well known [27,28,29], and both intraspecific, homologous and interspecific, non-homologous recombination are thought to be common and important in the evolution of poleroviruses. A few examples include BrYV [3], FBPV-1 [30], *Cucurbit aphid-borne yellows virus* (CABYV) [27,31], *Sugarcane yellow leaf virus* (ScYLV) [12,32], and *Cotton leafroll dwarf virus* (CLRDV) [33]. Our findings also suggest that recombination could play a role in the evolution of BrYV populations, as some of these genes, which are associated with compatible interactions, might be evolving under selection pressure from their hosts and/or vectors and tend to accumulate changes faster than other parts of the genome. Notably, mutations or recombination in genes can affect the biological functions of the viral proteins. However, how recombination could affect the biological functions associated with these genes remains a question worth studying in the future.

Our findings revealed that BrYV populations are evolving mainly under purifying selection pressure (Figure 5 and Figure 6 and Table 4). To gain an in-depth understanding of this selection factor at the gene level, we opted to estimate d_N_/d_S_ for P0, P1, P1-P2, P3, P3-P5, P4, and P5 genes of BrYV. While our results showed that the majority of the codons remained under negative selection for each gene with an average d_N_/d_S_ ratio of <1 (Figure 5 and Table 4), the overall contribution of negatively selected sites remained more than the positively selected sites (Figure 6). Since all the sites had both positive and negative selection pressures, this provided strong evidence of heterogeneous selection pressures among codon sites. This is in accordance with a previous study that concluded that the genes of the TuYV population in the UK were evolving under negative selection pressure [34].

Future studies focusing on the patterns of BrYV geographical distribution, evolutionary dynamics, and functional analysis of the BrYV genes arising from genetic variations will help us to understand the host (plant/insect)-BrYV relationships and viral pathogenicity in the context of developing strategies for sustainable virus disease management.

## 4. Materials and Methods

### 4.1. Sample Collection and Nucleic Acid Extraction

In November 2019, ten samples of wild radish (*R. raphanistrum*) were randomly collected at a commercial vegetable farm in Deloraine as part of an extensive survey across Tasmania to determine the population diversity of *Polerovirus* species in pea crops and the weed species present in the vicinity of the pea crops. All leaf samples were grouped into lots of five and preserved (at −80 °C) for subsequent testing. Extraction of total RNA was carried out using PowerLyzer 24 homogeniser (Qiagen, Chadstone, VIC, Australia) and RNeasy Plant Mini Kit (Qiagen, Hilden, Germany) according to the manufacturer’s instructions. Samples of 100 mg/lot were subject to RNA extraction to obtain sufficient yields of RNA for further processing and subsequent detection of viruses with low titer.

### 4.2. Next-Generation Sequencing (NGS) and Sequence Reads Analysis

Total RNA of the grouped sample 148 W was then sent to the Department of Jobs, Precincts and Regions, AgriBio, Bundoora, Victoria, for NGS. Libraries for RNA-Seq were prepared with a TruSeq Stranded Total RNA Sample Preparation Kit with Ribozero following the manufacturer’s instructions. To determine the size distribution and concentration of the RNA-Seq libraries, 2200 TapeStation^®^ system (Agilent Technologies, Santa Clara, CA, USA) and Qubit^®^ Fluorometer 2.0 (Invitrogen) were used, respectively. Afterwards, the Illumina NovaSeq was employed to sequence these libraries with a paired read length of 2 × 151 bp.

Trimming of the primer and adaptor sequences from raw reads was carried out using Geneious Prime 2021.2 (Biomatters Ltd., Auckland, New Zealand). For this purpose, the BBDuK plugin (provided in the BBTools package) was employed [35]. After that, the Velvet plugin (in Geneious Prime software) with default settings was used to perform de novo assembly. Then, a length-based sorting of the obtained contigs was carried out and, subsequently, blast (BLASTn and BLASTx) analysed against the GenBank database. Simultaneously, the mapping of contigs and reads to a collection of polerovirus reference genomes (available in GenBank) was carried out and results were also analysed in Geneious Prime.

### 4.3. Validation of NGS Results and Amplification of BrYV Genome

To synthesize the cDNA, an iScript Reverse Transcription Supermix Kit (BioRad Bio-Rad, Hercules, CA, USA) was used according to the provided instructions. The reaction mixture of 20 µL contained total RNA (2.5 µL), 5 × RT Supermix (4 µL), diethyl pyrocarbonate (DEPC), and treated water to make up the volume. The reaction was then incubated for priming at 22 °C (5 min), reverse transcription at 46 °C (20 min), and finally enzyme inactivation at 95 °C (1 min).

Several overlapping primer sets [17,18] were used to amplify the missing part of the virus genome, with reference to previously reported sequences of BrYV and TuYV (Table 1). This was done by RT-PCR-based amplification of the target sequence. The details, including primer sequences, amplicon size, target, and thermocycling conditions, are provided in Table 1. The assays were performed as follows: 1 × HotStarTaq Master Mix (Qiagen), 10 μM of each primer pair, 1 μL cDNA template, and made up to a total volume of 20 μL with sterile distilled water. Amplicon of appropriate sizes were then excised and purified using a QIAquick Gel Extraction Kit Protocol (Qiagen). Elution was carried out using a 30 μL elution buffer (10 mM Tris HCl, pH 8.5) and sent for bidirectional (sanger) sequencing by the Central Science Laboratories (CSL), University of Tasmania, Australia. The near full-length genomic sequence of BrYV Tasmanian isolates (BrYV-Tas) was then deposited in the NCBI GenBank under accession number OM469309. In addition, multiplex RT-PCR with specific primer sets (Table 1) was also performed to determine the genotype of BrYV, as described previously [19].

### 4.4. Multiple Sequence Alignment and Phylogenetic Analysis

To prepare multiple sequence alignments, the near-full-length genome sequence of BrYV-Tas was aligned with the full genome sequences of all reported BrYV isolates and one TuYV isolate (NC_003743), available in the GenBank (Table 2), using the MUSCLE option in Geneious Prime 2021.2. Similarly, all genes including P0, P1, P1-P2, P3, P3-P5, P4, and P5 were also aligned with the corresponding genes of the reported isolates, as mentioned above. Maximum likelihood phylogenetic trees were constructed in MEGAX with 1000 bootstrap replicates [36].

### 4.5. Recombination Analysis

The aligned genome of the BrYV isolate from this study and other BrYV isolates available in GenBank were tested for potential recombination events. The recombination analysis was carried out using RDP [37], GENECONV [38], BOOTSCAN [39], MAXCHI [40], CHIMAERA [41], SISCAN [42], and 3SEQ [43] methods implemented in the recombination detection program v.4 [44]. Aligned sequences were subjected to the above-mentioned methods using default parameters. Potential recombination events detected were considered statistically significant if supported by at least four programs with *p*-values lower than a Bonferroni-corrected cutoff of 0.05.

### 4.6. Nucleotide Diversity and Haplotype Variability Indices

The average pairwise number of nucleotide differences per site (nucleotide diversity, π) was detected using DnaSP v.5 [45]. The statistically significant differences among the mean nucleotide diversity from the datasets were determined by calculating their 95% bootstrap confidence intervals. In addition, the nucleotide diversity was calculated using a 100-nucleotide sliding window, with a 10-nucleotide step size. The number of segregating sites (S), the number of haplotypes (H), and haplotype diversity (H_d_) were also calculated for all datasets using DnaSP v.5 [45].

### 4.7. Detection of Positive and Negative Selection Sites

Using four different approaches, i.e., single-likelihood ancestor counting (SLAC), fixed-effects likelihood, random-effects likelihood, and partitioning for robust inference of selection [46], potential negatively and positively selected sites in the coding regions of P0, P1, P1-P2, P3, P3-P5, P4, and P5 were identified. All the analyses were performed in the Datamonkey web server (www.datamonkey.org accessed on 22 October 2021) [47]. Genetic Algorithm Recombination Detection (GARD) [48] was implemented to look for recombination breakpoints in the datasets in order to avoid ambiguous results. The dN/dS ratios were estimated using the SLAC method based on inferred GARD-corrected phylogenetic trees to compare the selection pressures acting on all genes of BrYV.

## Figures and Tables

**Figure 1 plants-11-00884-f001:**
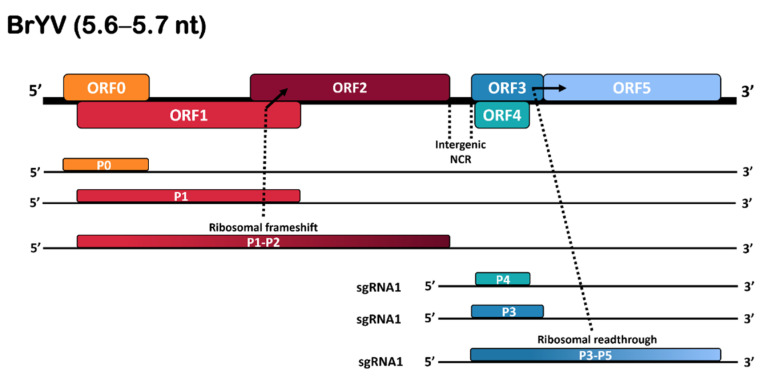
Representation of genomic organisation of BrYV. The ribosomal frameshift and read-through strategies are indicated by arrows; coloured boxes indicate the positions and types of different ORFs and translated proteins.

**Figure 3 plants-11-00884-f003:**
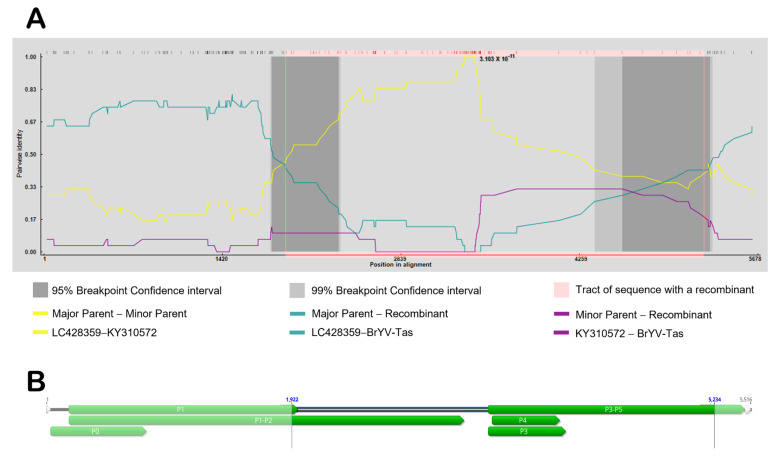
(**A**) Graphical representation of the recombinant region (pink). Pairwise identities of each pair of sequences (y-axis) and their position (x-axis) are indicated. (**B**) Schematic representation of BrYV-Tas isolate genome with the pattern of detected recombination breakpoints distributed in genome (highlighted).

**Figure 4 plants-11-00884-f004:**
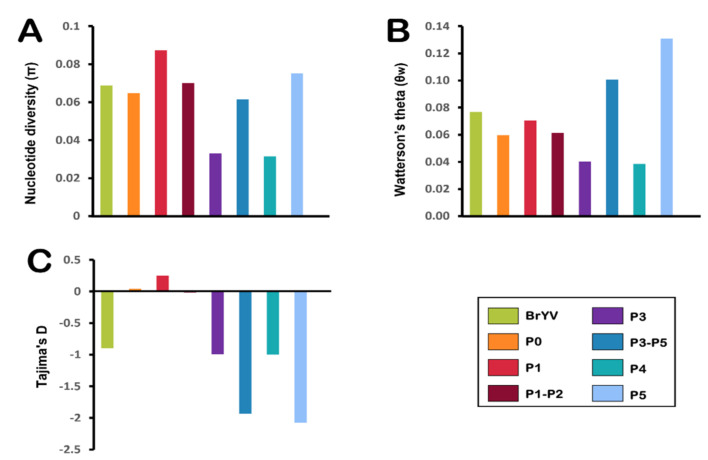
Estimation of genetic diversity was performed for the BrYV genome and individual genes. The calculated population genetic parameters include (**A**) nucleotide diversity (π), (**B**) Watterson’s theta (*θw*), and (**C**) Tajima’s *D*.

**Figure 5 plants-11-00884-f005:**
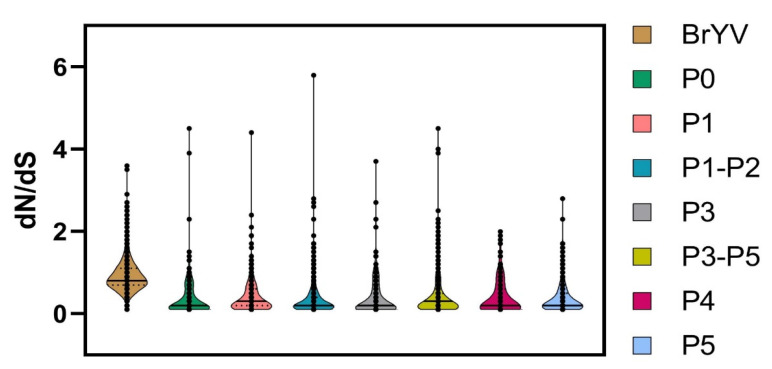
Estimation of selection pressure was performed by calculation of non-synonymous to synonymous substitution ratios (dN/dS). Violin plots correspond to dN/dS ratio for BrYV genome and its individual encoded genes. Solid horizontal lines represent median values, while dotted lines denote the lower and upper quartiles in each dataset.

**Figure 6 plants-11-00884-f006:**
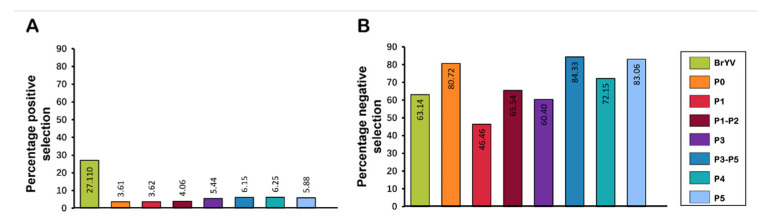
Percentage of sites evolving under negative/positive selection pressure. The datasets include (**A**) percentage of sites evolving under positive selection pressure in BrYV genome and its encoded genes and (**B**) percentage of negatively selected sites in BrYV genome and its encoded genes.

**Table 1 plants-11-00884-t001:** Primers used for RT-PCR, multiplex RT-PCR, and sequencing.

Primer Name	Sequence 5′ to 3′	Target	Position	Product Size (bp)	PCR Conditions	References
BrY-Ntab001F (+)	ACAAAAGAAACCAGGAGGRAA	BrYV ORF0	1	780	1 × 95 °C (15 min), 35 × [94 °C (30 s), 55 °C (30 s), 72 °C (1 min)], 1 × 72 °C (10 min)	[17,18]
TuYVOrf0R	TCATACAAACATTTCGGTGTAGAC		781			
BrY-Ntab724F (+)	TCTCACTCCTGAAGAAATCC	BrYV ORF1-ORF2	724	1349	1 × 95 °C (15 min), 35 × [94 °C (30 s), 55 °C (30 s), 72 °C (1 min)], 1 × 72 °C (10 min)	[17]
BrY-Ntab2064R (−)	TGAATCACACGCTCCCTCTCAG		2073			
BrYA484F	TACTTGGACTAGAGATGCTGAAAG	BrYV-ORF0 (Genotype A)	484	277	1 × 95 °C (10 min), 32 × [94 °C (30 s), 62 °C (60 s), 72°C (1 min)], 1 × 72 °C (10 min)	[19]
BrYB88F	CCTCCACCCAAAACAAGTAT	BrYV-ORF0 (Genotype B)	88	673		
BrYC257F	CGAGTTTCCGTACTTGTTG	BrYV-ORF0 (Genotype C)	257	504		
BrY761R	AGACCGAAGAGCTGAAAAGG	BrYV-ORF0 (Reverse)	761	-		

**Table 3 plants-11-00884-t003:** Molecular diversity among BrYV populations and individual genes.

Dataset	Number of Sequences	Total Number of Sites	S	Eta	H	Hd	π	*θw*	Tajima’s D
BrYV	20	5827	1477	1684	19	0.995	0.06883	0.07683	−0.89807
P0	20	751	159	171	19	0.995	0.06490	0.05976	0.04076
P1	20	1835	451	528	19	0.995	0.08736	0.07039	0.25116
P1-P2	20	3119	652	747	17	0.993	0.07001	0.06137	−0.01833
P3	20	609	87	94	18	0.989	0.03297	0.04027	−0.99362
P3-P5	20	2137	688	780	19	0.995	0.06144	0.10058	−1.93078
P4	20	528	72	78	14	0.932	0.03147	0.03844	−0.99753
P5	20	1342	506	578	19	0.995	0.07522	0.13073	−2.07541

S, number of polymorphic (segregating) sites; Eta, total number of mutations; H, number of haplotypes; Hd, haplotype diversity; π, nucleotide diversity; *θw*, Watterson’s theta.

**Table 4 plants-11-00884-t004:** Estimation of average dN/dS ratios and positive and negative selection for coding sequences of BrYV.

ORF	Avg. dN/dS Ratio	Positive Selection				Negative Selection			
		Total Sites	Avg.	Min.	Max.	Total Sites	Avg.	Min.	Max.
BrYV	0.902	501	1.341	1.051	3.613	1167	0.699	0.051	0.949
P0	0.376	9	2.040	1.050	4.54	201	0.286	0.054	0.935
P1	0.414	22	1.548	1.063	4.401	282	0.319	0.050	0.942
P1-P2	0.369	42	1.534	1.051	5.829	679	0.289	0.050	0.937
P3	0.409	11	1.777	1.084	3.660	122	0.265	0.050	0.918
P3-P5	0.409	42	1.775	1.060	4.462	576	0.294	0.050	0.942
P4	0.402	11	1.487	1.098	2.026	127	0.280	0.052	0.932
P5	0.371	25	1.459	1.058	2.778	353	0.288	0.050	0.949

## Data Availability

The data is contained within the article or Appendix A. The genome sequence of BrYV-Tas has been deposited in the NCBI database with Accession no: OM469309.

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
