# Peer review of "Genomic Characterisation of an Isolate of Brassica Yellows Virus Associated with Brassica Weed in Tasmania"

_plants, 2022, doi:10.3390/plants11070884_

Round 1
Reviewer 1 Report
The manuscript is well divided in the different paragraphs. Indeed, some general revision should be done to fulfill the requirements to be accepted. Please to uniform the reference style according to “Plants” guidelines.
There are some critical points about methodology and data description.
Please to find the comments reported in the pdf file attached.

Reviewer 2 Report
It was a pleasure reviewing the article entitled “Genomic characterisation of an isolate of Brassica yellows virus associated with brassica weed in Tasmania” submitted for consideration in the Plants. In the current study, a nearly complete genome sequence of BrYV (genotype A) isolated from Raphanus raphanistrum in Tasmania was described using next-generation sequencing and RT-PCR. The virus was identified as BrYV (genotype A) based on the nucleotide. Recombination analysis tools helped determine the major and minor parents of this isolated. This manuscript requires diligence revision before considering further. For instance, in the abstract and main text, it reads the newly identified BrYV-Tas (Submission ID 2545285) possesses a genome of 5516 nucleotides (nt) and shares”. Unfortunately, there is no clue where it has been submitted. Additionally, Line 113 states the length of BrYV-Tas as 5516-nt, whereas Line 253 mentions as 5522-nt.
To determine the nucleotide sequence of the 5’ end, RT-PCR was employed using primers as described in table 1. Further based on NGS, authors determined 1561 to 5505-nt (compared to BrYV isolate “TO3”). There is no description of how the nucleotide sequence between 5505 and 5516 (or 5522) was determined. Similar errors are found throughout.
Round 2
Reviewer 1 Report
Dear all thanks for the effort and the work done to revise the manuscript. I observed that some amendments were included in the text according to the comments raised, nevertheless I cannot be able to find a proper reply to all the comments included in the revised pdf file attached in the previous revision, or probably I’m missing the reply letter to my comments.
Some comments were completely ignored and not replied.
In particular:
Introduction:
Line 38: please to refer to study regarding ORF after that the genome features of BrYV were described.
Comment to section 2.1: “The comment was: the information included should be implemented. Please the author justify how only 670 reads were selected and on the basis of which criteria. there are data regarding N50 value and about the coverage. 670 reads to produce 16 contigs are really too few. It should included as supplementary material how reads generated each contigs and a maps about the distribution of the reads along the selected conting to see if some parts of the contings are not enough covered. in that case additional sanger sequecing should be done to cover the missing parts”
In the supplementary material was included a figure with contigs mappings, reporting that RT-PCR sequencing was done to obtain sequence in missing part. In the manuscript there is no reply to quality about the reads (values as N50 or the coverage). In my opinion 670 reads to produce 16 contigs are too small. The coverage values is crucial to understand if the contings were generate by a single reads or not. In the figure some genome regions were obtained by a single contigs. Even in this case some data regarding RT-PCR sequencing must be provided as the risk to introduce chimeric sequences/artefacts or bias is too high.
Comments to line 105:
“Why was not used a RACE assay to determine 5'/3' ends?”
Please to reply to this comment
Comment to line 368
“The amplicon were cloned prior to be sanger sequenced?”
Please to reply to this comment
Comment line 386
“It is not clear if the recombination analysis was done including only several BrYV isolates. The polerovirus genus is known to be characterized by occurring of recombination and reassortment of well distinct genome fragments.
in view of above, the recombination analysis of the full genome obtained should be done including other polerovirus species and not only BrYV.“
Now it is clear that the recombination analysis was done including only other BrVY. By the way could please the authors motivate why other full genome sequences of close related polerovirus were not included?
Reviewer 2 Report
It was a pleasure reviewing the revised version of the article entitled “Genomic characterisation of an isolate of Brassica yellows virus associated with brassica weed in Tasmania” submitted for consideration in the Plants. I am still not convinced about the revision. It needs more diligent revision. Below are a couple of examples.
- GenBank Accession no. OM469309 is not available in the databank.
- Table 1: amplicon size calculation (2079-725 = 1354 nt); Same for other primers too.
- Figure 2: Indicate the virus isolated in the present study for each gene
Round 3
Reviewer 1 Report
The manuscript has been improved and my doubts cleared, the manuscript can be published with some minor modifications mainly addressed to English language revision.
Author Response
We truly appreciate the reviewer’s encouraging remarks and constructive comments on the manuscript. The manuscript has been carefully revised by a native English scientist and needful grammatical, structural and typographical mistakes have been corrected to improve the readability. Also we are providing some additional supplementary data.
Reviewer 2 Report
None
Author Response

(The authors gave the same response as above.)
